# Towards Healthy Longevity: Comprehensive Insights from Molecular Targets and Biomarkers to Biological Clocks

**DOI:** 10.3390/ijms25126793

**Published:** 2024-06-20

**Authors:** Khalishah Yusri, Sanjay Kumar, Sheng Fong, Jan Gruber, Vincenzo Sorrentino

**Affiliations:** 1Department of Biochemistry, Yong Loo Lin School of Medicine, National University of Singapore, Singapore 117596, Singapore; 2Healthy Longevity Translational Research Program, Yong Loo Lin School of Medicine, National University of Singapore, Singapore 119228, Singapore; 3Department of Geriatric Medicine, Singapore General Hospital, Singapore 169608, Singapore; 4Clinical and Translational Sciences PhD Program, Duke-NUS Medical School, Singapore 169857, Singapore; 5Science Division, Yale-NUS College, Singapore 138527, Singapore; 6Department of Medical Biochemistry, Amsterdam UMC, Amsterdam Gastroenterology Endocrinology Metabolism and Amsterdam Neuroscience Cellular & Molecular Mechanisms, University of Amsterdam, Meibergdreef 9, 1105AZ Amsterdam, The Netherlands

**Keywords:** aging, biomarkers, biological clocks, geroprotector, supplements, drug development, clinical trials, omics, machine learning, healthy longevity

## Abstract

Aging is a complex and time-dependent decline in physiological function that affects most organisms, leading to increased risk of age-related diseases. Investigating the molecular underpinnings of aging is crucial to identify geroprotectors, precisely quantify biological age, and propose healthy longevity approaches. This review explores pathways that are currently being investigated as intervention targets and aging biomarkers spanning molecular, cellular, and systemic dimensions. Interventions that target these hallmarks may ameliorate the aging process, with some progressing to clinical trials. Biomarkers of these hallmarks are used to estimate biological aging and risk of aging-associated disease. Utilizing aging biomarkers, biological aging clocks can be constructed that predict a state of abnormal aging, age-related diseases, and increased mortality. Biological age estimation can therefore provide the basis for a fine-grained risk stratification by predicting all-cause mortality well ahead of the onset of specific diseases, thus offering a window for intervention. Yet, despite technological advancements, challenges persist due to individual variability and the dynamic nature of these biomarkers. Addressing this requires longitudinal studies for robust biomarker identification. Overall, utilizing the hallmarks of aging to discover new drug targets and develop new biomarkers opens new frontiers in medicine. Prospects involve multi-omics integration, machine learning, and personalized approaches for targeted interventions, promising a healthier aging population.

## 1. Introduction

Aging is characterized by a multifaceted decline in physiological function and stress resilience, ultimately leading to an increased risk of age-related diseases. While it is a complex biological process, several hallmarks of aging have been identified as a set of interconnected cellular processes that contribute to the aging phenotypes, including genomic instability, telomere attrition, epigenetic alterations, loss of proteostasis, deregulated nutrient sensing, mitochondrial dysfunction, cellular senescence, stem cell exhaustion, and altered intercellular communication [1] (Figure 1). Targeting these hallmarks, especially by means of lifestyle interventions, drugs and geroprotectors, has emerged as a promising strategy for regulating aging and promoting healthy longevity [2].

The investigation of aging hallmarks also aids in identifying biomarkers that reflect biological age. Biological aging reflects the health status of tissue and organ function, whereas chronological aging refers to the amount of time lived after birth. Individuals age at different rates from a biological perspective, hence it is important to monitor the risk of aging-associated disease using biomarkers. In fact, chronological age rarely corresponds to biological age, especially as organs also age at different paces [3]; therefore, biological age is a better reflection of risk of frailty and aging disease [4] (Figure 2).

Several biomarkers based on the hallmarks of aging are currently in development, which are further discussed in this review. For example, the measurement of telomere attrition and epigenetic alterations such as DNA methylation are popular biomarkers that measure aging [1,5]. Alternatively, cellular senescence markers can be measured in the blood and are predictive of biological age and age-related diseases [6]. Biomarker development facilitates personalized medicine by stratifying individuals based on an estimation of the biological age, paving the way for tailored interventions and evaluating the effectiveness of these interventions [1,7,8,9]. In addition, this allows optimization of treatment efficacy while minimizing adverse effects [10]. Moreover, biomarkers support epidemiological studies, enabling the investigation of aging-related risk factors and determinants [1].

Utilizing a combination of biomarkers including clinical parameters [11,12,13,14,15,16,17,18,19,20,21,22], DNA methylation [7,23,24,25,26,27,28,29,30,31], and biological -omics data [32,33,34,35,36,37,38,39], a variety of different biological aging clocks have been constructed. The goal of an aging clock is to provide an accurate readout of an individual’s true biological age, which indicates the person’s future risk of dying from all intrinsic causes of mortality and reflects that individual’s underlying biological resilience. Aging clocks should predict all-cause mortality well ahead of the onset of specific age-related diseases, thereby offering a window for early intervention. Yet, most current aging clocks do not delineate the aging mechanisms or identify the specific modifiable clinical drivers for biological aging and age-related diseases. At present, the development of such clocks is fraught with many challenges as discussed in this review.

A multitude of drugs, supplements, and lifestyle interventions have been developed to target aging in preclinical studies, with many advancing to clinical trials. Although supplements are widely in use to target aging such as nicotinamide adenine dinucleotide (NAD^+^) boosters and other vitamins, there is little evidence of improvement in clinical trials focused on aging. On the other hand, drugs that are able to slow the progression of aging preclinically, termed geroprotectors [40], are also under clinical translation, and these include molecules that can be repurposed to target aging pathways. One of the most established is the well-known geroprotector rapamycin, originally used as an immunosuppressant, that has promising results in multiple trials in the healthy elderly for metabolism, proteostasis, senescence, and immune response [41]. The aging hallmarks are intertwined, for example, cellular senescence can be caused by genomic instability and DNA damage [42]. Therefore, some clinically approved drugs can ameliorate multiple aging hallmarks by affecting a molecular target that may be common to two or multiple aging-related pathways. For example, rapamycin inhibits the nutrient sensing protein mechanistic target of rapamycin (mTOR), which is also implicated in the modulation of cellular senescence and autophagy [43,44,45]. Similarly, metformin, a widely used FDA-approved drug for diabetic patients, activates AMP-activated protein kinase (AMPK) which subsequently also inhibits mTOR and increases lifespan in mouse models [46].

A summary of the latest results on the effectiveness of drugs and supplements targeting the aging hallmarks in aging clinical trials is provided in this review. It also evaluates aging biomarkers currently in use along with prospective biomarkers. Although a large number of clinical trials show efficiency of calorie restriction and physical activity on aging [47,48], this review focuses on molecular targets of aging that have measurable biomarkers associated with the hallmarks. Anti-aging strategies such as stem cell therapy and microbiome modulation aim at achieving broader therapeutic effects spanning across different biological domains, not always measurable via specific dedicated biomarkers and are therefore not a primary focus of this review. Readers are invited to explore these topics as reviewed by Zhu et al. [49] and Badal et al. [50]. Similarly, inflammaging, or age-related chronic inflammation, is a major component of aging and age-related diseases with a myriad of contributing factors ranging from cellular senescence to gut microbiome dysfunction [51]. While inflammatory markers themselves are useful biomarkers of aging [52], there is no specific target against inflammaging, with many proposed therapeutics being geroprotectors known for their anti-inflammatory properties such as rapamycin and metformin [53]. However, senolytic drugs and NAD^+^ boosters are also suggested to be effective thanks to their broad-spectrum beneficial effects due to the interconnectivity of the hallmarks [53,54]. Oxidative stress, another critical alteration of cellular and tissular homeostatic processes, was defined by Harman as an accumulation of free radicals during the aging process, which results in damage to cellular components, thus causing aging phenotypes [55,56]. Oxidative stress may also be considered a hallmark of aging; however, due to the broadness of its impact on all the aging hallmarks as well as inflammaging [57], measuring the efficiency of targeting oxidative stress to modulate free radical levels essentially amounts to measuring the downstream effects on readouts of other hallmarks. Finally, this review summarizes the state-of-the-art on the development and use of biological aging clocks and highlights the limitations and possible steps forward in this constantly evolving field. Tools to accurately quantify changes in true biological age are essential for the development and validation of any healthspan- and lifespan-optimizing diet, lifestyle, supplement, or drug intervention.

## 2. Current Targets of Aging Interventions

A variety of compounds are currently under investigation or have recently been studied for geroprotector effects in clinical trials, listed in Table 1. These involve either natural compounds, often used under the label of supplementation approaches, or drug-based interventions, often based on their repurposing from disease applications to the field of healthy aging. With the progress in characterization of the biological mechanisms of drugs, disease and aging pathways, overlapping molecular targets are emerging, which may play pivotal roles in the context of healthy longevity strategies, as summarized below.

### 2.1. Cellular Senescence

Senescent cells accumulate with aging in several organs and are a major cause of age-associated disease [108,109,110,111]. Senescence is an irreversible form of long-term cell-cycle arrest induced by excessive intracellular or extracellular stress or damage. Senescent cells exhibit distinct molecular and morphological features, such as the expression of senescence markers and an enlarged appearance [112,113]. There are two primary types of senescence processes: replicative senescence, caused by telomere shortening, and stress-induced premature senescence, triggered by various stressors including DNA damage, oncogene activation, oxidative stress, chemotherapy, mitochondrial dysfunction, epigenetic changes, and paracrine signaling [112,113,114,115,116]. Senescent cells, though arrested, remain metabolically active, undergoing changes in degradation pathways and energy generation. The purpose of this arrest is to limit the proliferation of damaged cells and prevent malignant transformation [117,118]. However, during aging senescence, an unfavorable response to cellular stress and genome instability occurs. Cellular senescence encompasses two temporal categories: acute, which is transient and beneficial in normal biological processes such as tissue repair, and chronic, induced by prolonged stress or damage. Chronic senescence, characterized by high resistance to immune clearance, has detrimental effects on cells and tissues due to its prolonged nature and distinct effector pathways [119,120,121,122]. DNA damage can result from defective DNA damage response and repair mechanisms, aberrant DNA replication, and cellular stressors such as reactive oxygen species (ROS) and other genotoxins [123,124]. The DNA damage response arrests cellular replication to allow for DNA repair, resulting in genome instability, which induces cellular senescence [125,126]. Paradoxically, activation of oncogenes can induce senescence despite tumor suppressor genes being markers for senescence. This occurs as the initial increase in DNA replication upon oncogene activation results in activation of the DNA damage response [127]. Another event linked to cellular senescence is telomere shortening. Telomeres are repeated DNA sequences at the ends of chromosomes that decrease in length with age [128]. During cellular division, the DNA replication machinery is unable to replicate the entire DNA molecule, resulting in a shortened telomere. Telomerase increases telomere length by adding repeat sequences to the 3′ end. Most somatic cells lack telomerase activity, whereas germ and stem cells are high in telomerase activity [129]. When telomeres are shortened to a critical length, replication is arrested, and cell senescence occurs [130].

The signatures of senescent cells are an accumulation of senescence-associated β-galactosidase (SA-β-gal) and cell cycle inhibitors, notably p16^INK4A^, p21^CIP1^, and p53, hence halting cellular proliferation. Senescent cells are resistant to apoptosis through an increase in antiapoptotic protein BCL-2 [131]. p16 is a cyclin-dependent kinase (CDK) inhibitor that mediates transition from the G1 to S phase by hypophosphorylating retinoblastoma protein. The RAS oncogene pathway can induce p16 expression, conferring oncogene-induced senescence [132]. The transcription factor p53 is induced by DNA damage, which increases p21CIP1 expression, a CDK inhibitor that arrests cells at the G2 phase [133]. The senescence-associated secretory phenotype (SASP) comprises factors secreted by senescent cells to reinforce or to induce senescence in surrounding cells. SASP factors include proinflammatory cytokines interleukin-6 (IL-6) and IL-8 and are an attractive drug target. The SASP triggers tissue dysfunction through induction of chronic inflammation that causes age-associated diseases [134]. Therefore, targeting the different components of cellular senescence such as the SASP and telomere shortening could prevent age-related disease. Failure to clear senescent cells in aging leads to tissue damage and fibrosis, highlighting the importance of developing these therapeutics [135].

*Senolytics—*Senolytics are a category of compounds that selectively eliminate senescent cells. Dasatinib is an FDA-approved kinase inhibitor for the treatment of chronic myeloid leukemia. Preclinical studies show that the mechanism of dasatinib is apoptotic induction in senescent cells through inhibition of the Src family kinases [136]. Quercetin is a natural plant flavonoid consumed as a supplement and is generally recognized as safe (GRAS). It induces apoptosis in multiple cancer cell types through a decrease in survivin and Bcl-2 and activation of p53 [137]. Dasatinib and quercetin were first combined as each drug was effective against different cell lines, and the combination was able to eliminate a broader range of senescent cell types than either drug alone [138]. In a long-term trial of nonhuman primates, dasatinib and quercetin reduced senescence markers p16 and p21 expression in adipose tissues [139]. At the clinical level, a combination of dasatinib and quercetin for 3 days reduced p16 and p21 expression in the skin of elderly subjects and reduced circulating SASP factors [58]. Epigenetic age was measured in aged subjects administered dasatinib and quercetin for 3 to 6 months [59]. Although epigenetic age increased for first- and second-generation clocks, there was no significant change in epigenetic age for one second generation clock and third generation clocks, suggesting reduced biological aging.

Fisetin, a natural flavonoid, reduces cellular senescence in cardiac and white adipose tissue and stem and immune cells of old mice [62], in higher mammals in the sheep brain, liver, and lungs [64], and in aged human skin grafts on mice [140]. Fisetin supplementation was also able to increase lifespan in mice [63]. Cellular experiments show fisetin acts on senescence through activation of phosphatase and tensin homolog (PTEN), which in turn downregulates the mTOR2-Akt pathway in vascular smooth muscle cells [141]. Overall, preclinical studies show evidence for the use of fisetin as a senolytic, with prospects moving into human trials [142].

*Senomorphics*—senomorphics are drugs that inhibit components of the SASP to reduce senescence without killing the cells [143]. NF-kB signaling, a master regulator of inflammatory cytokines, can be inhibited by small molecules such as SR12343, which is able to reduce senescent markers in mouse models of accelerated aging [144]. The Janus kinase/signal transducer and activator of transcription (JAK/STAT) pathway regulates cytokine production and drives the SASP. Ruxolitinib, an FDA-approved drug for myelofibrosis and a JAK inhibitor, decreases inflammatory cytokines in aged mice [65].

*Geroprotectors*—Although geroprotectors do not specifically target senescence, these drugs have beneficial impacts on limiting senescence. Rapamycin can inhibit cellular senescence and SASP through activation of the nuclear respiratory factor 2 (NRF2) and JAK/STAT pathway and reduced NF-κB transcriptional activity [145,146]. Topical rapamycin treatment for 6–8 months in subjects older than 40 years was able to reduce expression of p16, suggesting a reduction in senescence, along with improvements in skin photoaging [66]. Other clinical trials involving rapamycin in aged subjects are ongoing for general body function and condition, cardiac function, and ovarian aging.

*Telomerase activators—*Telomerase activators, such as GRN510 and nutraceutical cycloastragenol (TA-65), are prospective anti-aging compounds that may inhibit cellular senescence. As telomere shortening causes replicative senescence, telomerase activators could prevent senescence [147]. Indeed, elongating short telomeres in mice with TA-65 increased health span, although senescence was not measured [148]. In mouse studies, GRN510 treatment in young and old mice was able to improve motor function, with an increase in telomerase reverse transcriptase (TERT) in the brain and an increase in cognitive function in a Parkinson’s disease model [149,150]. However, telomere length did not increase, therefore effects of telomerase on improved brain function are most likely non-canonical. TA-65 was tested in elderly patients previously infected with cytomegalovirus. TA-65 increased telomere length in contrast to a placebo, which decreased telomere length over a 12-month period [70]. Another trial administered TA-65 in myocardial infarction patients aged over 65 and observed improvements in inflammation and lymphocyte populations after 12 months [71]. Telomere length was not measured. Recently, natural product 08AGTLF was identified as a more potent activator of telomerase; however, preclinical studies are lacking [151].

Overall, targeting cellular senescence to combat aging has good prospectives with senolytics entering clinical trials. Combining senolytic drugs or techniques to target various senescent cell types could enhance efficacy, as seen in the case of dasatinib and quercetin [138,152]. Gerotherapeutics that also inhibit senescence are widely studied in clinical trials. However, senomorphics are an under-studied area despite the importance of the SASP in aging senescence. Furthermore, combining senotherapeutics with other interventions targeting healthy aging could enhance treatment effects in age-related disorders due to the interconnected nature of cellular senescence and other hallmarks of aging.

### 2.2. Autophagy

Autophagy is a cellular process crucial for maintaining homeostasis by degrading and recycling damaged or unnecessary cellular components, such as defective organelles and aggregates of misfolded proteins, through lysosomes. There are three main types of autophagy: macroautophagy, microautophagy, and chaperone-mediated autophagy (CMA), each involving distinct mechanisms of cargo delivery to lysosomes for degradation. Macroautophagy involves the formation of double-membraned vesicles called autophagosomes, regulated by the ATG proteins, which engulf cytoplasmic material and fuse with lysosomes into autolysosomes for degradation [153]. Necessary for autophagosome formation is the conjugation of the cytosolic microtubule-associated protein 1 light chain 3 (LC3-I) with the phospholipid phosphatidylethanolamine to form LC3-II. LC3-II then localizes to the autophagic membrane; therefore, the dynamics of LC3-I and LC3-II are often used as markers of autophagy in preclinical studies [154]. In microautophagy, lysosomes or late endosomes directly engulf cytoplasmic material through invaginations or protrusions of the membrane, which then fuse into the endolysosome in which cargo is degraded [155]. Macro- and microautophagy can be either nonspecific, for the degradation of bulk cytoplasm, or highly specific, which targets specific organelles or proteins. Selective autophagy targets specific cytoplasmic cargo through the recognition of autophagy receptors that interact with LC3 in the autophagic membrane, leading to cargo sequestration into autophagosomes for lysosomal degradation. There are multiple methods of cargo recognition including ubiquitination, binding of sugars such as lectins, or of lipids such as cardiolipin [156]. Selective autophagy of mitochondria, termed mitophagy, is a quality control mechanism that degrades damaged mitochondria, the decline of which is associated with many age-related diseases, which will be further elaborated on in later sections. On the other hand, CMA is a selective form of autophagy that targets proteins containing a KFERQ-like motif that binds to cytosolic chaperone heat shock cognate 71 kDa protein (HSC70) [157]. The target protein is then imported into lysosomes through interaction with the lysosome-associated membrane protein type 2A (LAMP2A) receptor for degradation [158]. Microautophagy can also target HSC70-chaperoned proteins with the KFERQ-like motif; however, these proteins are invaginated in the lysosome, rather than transported through the LAMP2A receptor [158]. Autophagy is protective against aging diseases such as cancer and neurodegeneration. It also decreases with age in cells, brains of animal models, and the human brain [159,160]. Activation of autophagy pharmacologically or genetically in model organisms increases lifespan, which is abolished with the knockdown of autophagy genes [161], highlighting the importance of autophagy in longevity.

*Activators of autophagy*—Given the essential role of the different autophagic branches in the context of aging and age-associated ailments, research efforts are proposing several compound-based strategies to activate autophagy and its therapeutic benefits [159]. There are a variety of natural compounds able to induce autophagy that are not extensively studied. Trehalose, a disaccharide sugar, stimulates autophagy independently of mTOR by activating transcription factor EB (TFEB), a master regulator of lysosomal biogenesis and autophagy-related gene expression [162,163]. Preclinically, trehalose clears misfolded and aggregated proteins in cells and a mouse neurodegeneration model, reducing brain atrophy [164,165]. These results show good prospects for studying trehalose in human trials of aging-related neurodegenerative diseases.

Spermidine, a naturally occurring polyamine, has been shown to induce autophagy in multiple ways including inhibition of histone acetyltransferases and hypusination of eIF5A resulting in TFEB translation [166]. It is an essential metabolite regulating cellular functions through the polyamine pathway. Spermidine extends the lifespan of yeast, worms, flies, and mice by enhancing autophagy [167,168]. Several trials testing spermidine in the elderly on cognitive function have been performed. Short-term (3-month) trials show an improvement in cognition [74,75]; however, a long-term (12-month) trial showed no difference in memory [76]. Other trials of spermidine for elderly patients will begin in the context of coronary artery disease and SARS-CoV-2 vaccine efficiency.

*Geroprotectors—*Metformin can induce autophagy through inhibition of mitochondrial complex I, resulting in increased AMP/ATP ratios and decreased ATP levels. This results in AMPK activation, which inhibits mTOR and activates TFEB [169]. In preclinical studies, CD4+ T cells from older subjects contained lower autophagy markers compared to younger subjects. When treated with metformin, the T cells from older subjects increased in LC3-II expression suggesting increased autophagy, which was not observed in the young cells [77]. Mouse studies of rapamycin also show increases in autophagy [69]. These preclinical studies highlight autophagy as one of the mechanisms of geroprotection conferred by these pleiotropic drugs.

Overall, autophagy is central to aging and maintaining protein homeostasis, especially considering the importance of protein aggregates in age-associated neurodegenerative diseases such as Alzheimer’s (AD) and Parkinson’s disease (PD). Hence, natural compounds and geroprotectors targeting autophagy are an attractive avenue for longevity medicine. It is important to note that there are no therapeutics specific for the different forms of autophagy, i.e., macroautophagy, microautophagy, and CMA, which may be a worthy avenue of future research.

### 2.3. Protein Quality Control

Protein quality control (PQC) ensures proper protein synthesis, folding, trafficking, and degradation, crucial for maintaining cellular equilibrium and function. Dysregulation of PQC results in the accumulation of misfolded or aggregated proteins, hallmark features of age-related diseases such as AD and PD. An in-depth review of the role of PQC in aging is provided by [170]. With aging, there is a gradual decline in PQC, leading to the accumulation of damaged proteins and dysfunctional organelles, ultimately contributing to age-related diseases. This decline is evolutionarily-conserved, showing the criticality of the process in maintaining healthy aging [171]. Preclinical studies in *C. elegans* show a decline in protein synthesis during aging as ribosomal and transcriptional machinery content declines, resulting in a general protein imbalance [172,173]. Following translation, proper structure of proteins are ensured by the heat shock proteins (HSPs) that chaperone proteins for folding and refolding. HSPs are induced by the heat shock response (HSR) as a result of cellular stressors such as heat, oxidative stress, and heavy metals, preventing stress-induced unfolding of proteins. They also mediate protein degradation through the ubiquitin–proteasome system (UPS) [174].

Partially folded or misfolded proteins tend to form aggregates, which are associated with diseases such as neurodegeneration, highlighting the importance of removal. The HSPs are extensively studied, with some being more physiologically impactful than others. HSP70 is a major player in PQC that takes part in a wide range of processes from protein folding, refolding, translocation, disaggregation, and degradation [175]. HSP70 binds to hydrophobic regions on native proteins in an ATP-dependent manner, preventing them from aggregation and allowing the hydrophobic regions to bury [176]. It also acts as a chaperone for protein degradation through CMA and UPS [174]. HSP70 can interact with other chaperones and co-chaperones, one of them being the equally significant HSP90 [175]. HSP90 regulates final structural folding and undergoes structural changes upon ATP hydrolysis that aid protein folding [176]. The expression of HSP genes are induced during stress by heat shock transcription factors (HSF), which are in turn negatively regulated by the HSPs [176]. HSP expression and activity decline with age in human blood and brain, heightening susceptibility to protein misfolding and aggregation [177,178].

Post-translational modifications (PTMs) modulate protein stability, activity, and localization, thereby influencing proteostasis and cellular function [179]. Phosphorylation, ubiquitylation, and acetylation are essential PTMs directing proteins for autophagy and the UPS; therefore, dysregulation in PTM results in aggregation of aberrant proteins. For example, phosphorylation on sites in the tau protein are associated with clinical progression [180]. Both enzymatic and nonenzymatic PTMs are affected by age due to metabolic changes and ROS [179].

While autophagy is also part of the PCQ through mediating the clearance of protein aggregates and damaged organelles as discussed earlier, the ubiquitin–proteasome system (UPS) targets short-lived and misfolded proteins for degradation [181]. The proteasome consists of the 20S catalytic core and two 19S regulatory caps, that, combined together, form the 26S proteasome [182]. Proteins with and without a polyubiquitin chain can be degraded by the 20S and 26S ribosome [183,184]. Ubiquitination is achieved by the E1–3 ubiquitin ligases [182]. The UPS pathway exhibits decreased efficiency with age; proteasome expression and activity decrease during aging [185,186,187]. Proteasome disassembly also increases due to mitochondrial stress leading to ROS generation [188,189]. Conversely, preclinical studies in *C. elegans* show that hyperactivating the proteasome increases lifespan and stress response [190].

At the preclinical level, several natural products have been identified that increase chaperone expression, enhancing the UPR and UPS, as reviewed by Cuanalo-Contreras and Morena-Gonzalez [191]. In *C. elegans*, transcriptomics analysis identified monorden and tanespimycin, inhibitors of Hsp90 that increase expression of UPR genes [79]. The mechanism was found to operate through HSF1 activation due to the Hsp90 inhibition, which in turn allows transcription of UPR genes. Tanespimycin was previously studied in clinical trials for cancer [192]; however, trials have since been halted. The results by Janssens et al. [79] suggest that tanespimycin can be repurposed into a gerotherapeutic, and other inhibitors of Hsp90 like monorden are an attractive avenue of study [193]. Compounds that stabilize and activate the proteasome by improving substrate entry into the proteasome or allosteric agonists that enhance substrate binding are of particular interest to combat aging [194]. Candidate compounds for the activation of the proteasome are reviewed by Njomen and Tepe [195]. In particular, the phenothiazines, a class of drugs used to treat mental illnesses, can enhance 20S activity [196]. These drugs increase lifespan in *C. elegans* [80] and are able to cross the blood–brain barrier, providing a good basis for the development of further drugs. Providing omega-3 polyunsaturated fatty acid to older adults increased muscle protein synthesis [81], decreased expression of ubiquitin-mediated proteolysis pathways [82], and improved muscle function [83], suggesting supplementation approaches which prevent excessive protein degradation as a method to prevent muscle wasting. Overall, modulation of the PQC is as yet an under-explored target compared to the other hallmarks, with only preclinical studies showing improvement in aging.

### 2.4. Mitochondria

Central to many of the hallmarks discussed earlier is mitochondrial dysfunction [197]. As organisms age, mitochondrial function deteriorates, leading to increased oxidative stress, impaired energy production, and accumulation of mitochondrial DNA (mtDNA) mutations [198,199]. Mitochondrial dysfunction can also cause mitochondrial dysfunction-associated senescence through sirtuins and ROS [115,200]. ROS production within mitochondria is a natural consequence of cellular respiration, particularly during oxidative phosphorylation. The cells possess their own anti-oxidant system through enzymes such as superoxide dismutase and catalase, and free radical scavengers such as glutathione and vitamin E and C [201]. As humans age, cells become less effective in neutralizing ROS, leading to the accumulation of oxidative damage [202]. This corresponds to oxidative damage to proteins, lipids, and nucleic acids. Mitochondrial DNA (mtDNA) mutations occur due to various factors such as replication errors, exposure to environmental toxins, and oxidative damage from ROS [203]. The accumulation of mutations in mtDNA has been recognized as a significant contributor to aging and age-related diseases [204,205,206,207]. Aging humans show elevated levels of mitochondrial genome mutations in the brain, heart, and skeletal muscles compared to younger individuals [208,209,210]. During aging, a decline in mitochondrial respiratory capacity and ATP production were observed in aged human muscle biopsies, which leads to changes in cellular metabolism and energy homeostasis [211]. This can in turn affect other hallmarks such as PQC and autophagy [188,212]. Alterations in mitochondrial dynamics, including fusion, fission, and mitophagy, occur in the aging process [213]. Mitochondrial dynamics are tightly regulated by a network of proteins, including dynamin-related GTPases, such as mitofusins (MFN1 and MFN2) and atrophy 1 (OPA1) for fusion and dynamin-related protein 1 (DRP1) for fission. Moreover, mitochondrial biogenesis, the process by which new mitochondria are generated, declines with age, further exacerbating mitochondrial dysfunction [214]. The peroxisome proliferator-activated receptor gamma coactivator 1-alpha (PGC-1α) and nuclear respiratory factors (NRFs) are key regulators of mitochondrial biogenesis, whose expression decreases with age [215,216].

Aging can lead to impaired mitophagy in human muscle and heart [217,218] and in the onset of age-associated diseases [219]. Clearance of damaged mitochondria is essential for the prevention of mitochondrial dysfunction and cell death. Mitophagy is regulated by various pathways, including (PTEN)-induced putative kinase 1 PINK1 and the E3-ubiquitin ligase Parkin [220]. Alternatively, mitophagy can be induced independently of ubiquitination by BNIP3 in response to hypoxic conditions, which causes mitochondrial membrane depolarization and permeabilization and an increase in ROS [221]. Mitophagy can be activated via inhibition of mTOR or activation of AMPK [222]. Hence, mTOR inhibitors such as rapamycin also activate mitophagy [223].

*Mitophagy inducers*—The discovery of Urolithin A (UA) as a natural compound with mitophagy boosting properties has made a big impact on the longevity field, and the compound has quickly progressed into clinical trials and into the market as supplements. Preclinical studies show UA can increase mobility and lifespan in *C. elegans* in a mitochondrial-dependent manner. In the mouse, UA induces autophagy and mitophagy and increases muscle strength [84]. Stirred by these positive results in muscle function, clinical trials were launched. In a short-term trial in older adults, an increase in mitochondrial biogenesis was observed [85]. In a long-term 4-month trial, UA treatment increased muscle strength, along with significant increases in mitophagy observed by increased protein levels of Parkin and BNIP3 [86]. Other mitophagy inducers include oleanolic acid, which was able to increase mitophagy in aged mice [224], and the natural compound honokiol, which restores cognitive function in an Alzheimer’s mouse model with increases in mitophagy and fusion proteins MFN1-2 and OPA1 [225].

*Other mitochondrial therapies*—CoQ, or ubiquinone, functions as an electron carrier in the electron transport chain, a structural component of complex I and III, as well as an antioxidant to cell membranes and plasma lipoproteins. CoQ10 supplementation has been trialed in diseases such as cardiovascular disease, type 2 diabetes, and neurodegenerative diseases, and hence may have use in aging diseases [226]. In a trial of elderly women, CoQ10 supplementation showed benefits in the skin including increased collagen synthesis and elasticity [87]. MitoQ, a derivative of CoQ10, has entered the market as a supplement. Preclinical studies have shown its ability to increase mitochondrial membrane potential [227]. In human trials, MitoQ has been shown to improve vascular function in the elderly [88].

Currently, targeting of mtDNA mutations is being investigated using MitoTALENS, engineered nucleases encoded in a viral vector that target specific mtDNA mutations [89]. It is effective in mice and could be a method of elimination of mtDNA mutations in aging humans [228]. However, with all methods of gene editing, there is a long road to safety and ethical approval before testing MitoTALENS in human trials.

Epicatechin, a flavonoid that is available as a supplement on the market, is able to induce mitochondrial biogenesis [229]. In small human trials, epicatechin supplementation improved mitochondrial structure through an increase in cristae [90]. Small molecules aimed to improve mitochondrial dynamics via activation of Mfn1-2 are also sought after. S89 activates Mfn1, restoring mitochondria during oxidative stress in cells [230]. A molecule identified by Zacharioudakis et al. [231], labelled MASM7, is able to activate MFN1-2 and increase mitochondrial respiration and ATP production. Mdivi-1 inhibits DRP1 to inhibit mitochondrial fission [232]. Sulforophane, a dietary supplement, is an NRF2 activator that improves mitochondrial function and decreases oxidative stress in the aged rat kidney [91] and increases mitochondrial protein expression in the aged mice heart [92].

*Gerotherapeutics*—Metformin activates PGC-1a through AMPK, increasing mitochondrial biogenesis and fission [233]. In aged subjects on 6 weeks of metformin, several transcriptomic changes were observed, notably an improvement in DNA repair, mitochondrial pathways, and lipid metabolism [78]. Rapamycin supplementation in mice resulted in a reduction in mtDNA mutations, a reduction in ROS production through enhancement of complex I, and a reduction in the decline of mitochondrial function [67,68,234].

The biological processes of aging are interlinked and targeting of one hallmark of aging will likely benefit another. As mitochondria and oxidative stress are central to many aging pathologies, pleiotropic therapeutics such as rapamycin and metformin will positively affect all hallmarks of aging also through their influence on mitochondria.

### 2.5. NAD^+^

NAD^+^ is a crucial coenzyme concentrated in the mitochondria, involved in various cellular processes, including energy metabolism, DNA repair, and epigenetic regulation. NAD^+^ levels decline with age in some organs such as the brain and liver, as reviewed by Peluso et al. [235], leading to impaired cellular function and contributing to aging of certain tissues like skeletal muscle and age-related diseases such as sarcopenia and Werner’s syndrome, both preclinically and in humans [236,237,238]. NAD^+^ plays a central role in cellular metabolism, serving as a cofactor for enzymes involved in glycolysis, the tricarboxylic acid cycle, and oxidative phosphorylation which generate ATP through substrate-level phosphorylation [239]. Reduced NAD^+^ levels result in impaired mitochondrial activity and increased ROS generation [240]. As a majority of cellular processes require ATP, NAD^+^ decline would affect processes involved with the other aging hallmarks such as metabolic dysfunction, mitochondrial dysfunction, and autophagy. Additionally, NAD^+^ is a cofactor for hundreds of enzymes implicated in the aging process such as sirtuins, poly (ADP-ribose)-polymerases (PARPs), and CD38 [241]. The age-related decline in NAD^+^ has been hypothesized to be due to increased activity of NAD^+^-dependent enzymes, in particular PARPs, in response to DNA damage [241]. The molecule is in high demand and is constantly being synthesized de novo or from precursors and is recycled from the reduced molecule NADH [241]. NAD^+^ precursors can be endogenous metabolites or originate from the diet such as vitamin B3: nicotinamide riboside (NR), nicotinamide (NAM), and nicotinic acid (NA) [242]. Dietary NAD^+^ precursors NR, NA, NAM, and nicotinamide mononucleotide (NMN) are attractive supplements on the market that are undergoing investigation in clinical trials.

*NAD^+^ precursors*—Preclinically, NR improves proteostasis in muscle and the brain in mouse models and improves mitochondrial function in *C. elegans* [243,244]. In mice, NR improved lifespan, prevented stem cell senescence, and improved mitochondrial function [97]. Similarly, NMN, also a popular supplement, reduced oxidative stress and protein aggregation, and improved mitochondrial structure in aged mice [103]. In clinical trials, NR supplementation improved physical performance in healthy aged adults [93] and reduced inflammation and mitochondrial gene expression while maintaining mitochondrial bioenergetics [94]. However, trials in elderly with mild cognitive impairment showed no improvement in cognitive function [94]. In PD patients, NR shows decreased inflammation and a trend towards improved disease progression [95]. In healthy aged adults, NMN improved physical fitness [245,246], increased HDL, and decreased HbA1c levels [102]. Trials in elderly prediabetic women showed improved insulin sensitivity [99,100]. However, another trial showed no improvement in insulin resistance [99,100]. NMN treatment showed no increase in blood biological age during the treatment period while placebo biological age did increase, suggesting improvements in biological age [99]. NA has been studied clinically since 1955 for its lipid-lowering activity; however, aging trials are scarce due to the undesirable skin flushing side effect [247]. In the healthy elderly, NA was able to decrease insulin sensitivity, with declines in triglyceride and cholesterol levels [104], and improve memory in an old study from 1985 [105]. NAM is widely used in skincare to combat skin aging with good results [248]; however, there are no other aging trials for NAM to our knowledge.

Further evidence of the complexity of NAD^+^ metabolism and its pathophysiology in elderly humans also recently highlighted the possibility that other novel NAD^+^ precursors exist and may hold promise for healthy longevity applications, such as trigonelline, a methylated form of NA that also boosts NAD^+^ levels across species [106]. Trigonelline supplementation in preclinical models also showed benefits for age-associated cognitive decline and muscle function loss [106,249], but clinical studies remain to be developed.

*CD38 inhibitors*—CD38 is an NADase that modulates T cell function to produce cytokines [250]. It has been shown that CD38 inhibition by small molecules, or loss of expression in knockout mice, is able to prevent the age-dependent decline in NAD^+^ levels [251]. Multiple molecules have been developed or discovered for the inhibition of CD38 such as compound 1 from Li et al. [252], 78c [251], and apigenin [253], which have positively impacted the health span of mice. Quercetin has also been reported to act as a CD38 inhibitor [253]; hence, longevity-promoting activities of the molecule may occur also via this mechanism. Recently, Napa Therapeutics is developing a new molecule inhibitor of CD38, NTX-748, for rheumatoid arthritis, which has good results in mouse models and is able to increase NAD^+^ levels in the liver and spleen [107].

Collectively, the examples above indicate that NAD^+^ precursors and compounds that contribute to boost NAD^+^ are clearly one of the most currently investigated interventions for ameliorating the aging process and related disease; however, conclusive clinical evidence for these claims is still lacking and requires further testing with stringent clinical designs, larger cohorts, and possibly direct comparisons of the different bioactives.

## 3. Biomarker Identification and Validation Criteria

Investigating the hallmarks of aging from a molecular perspective is not only key to map targets and interventions as discussed in the previous chapters but also for the identification of aging biomarkers. There is currently no widely accepted single-measurement biomarker of aging, and it is unlikely to be discovered due to the complexity of the aging process. Ideal biomarkers of aging proposed over the past decades should meet several criteria. Firstly, they should be minimally invasive and reliable, allowing for longitudinal measurements with low technical variability. Secondly, biomarkers should be relevant to the aging process itself. Thirdly, they should predict the physiological aspects of aging, such as mortality, more accurately than chronological age. Lastly, they should demonstrate responsiveness to longevity interventions [254,255,256,257]. More precisely, assessment criteria include feasibility, validity, age-sensitivity, mechanistic significance, generalizability, response, and cost implications. Feasibility includes non-lethality to model animals and minimal invasiveness to humans, repeatability, and short timeframe relative to the organism’s lifespan [254,258]. Age sensitivity criteria for identifying biomarkers of aging require the biomarker’s level increase with chronological age, reflecting true biological aging. Such biomarkers able to differentiate between chronological and biological age are quantified as “age deviation” (AgeDev). To ensure precision, age-sensitive biomarkers are designed to account for chronological age, thereby strengthening their association with AgeDev and improving their capability to track biological aging effectively [23,259,260]. Mechanistic aging criteria refer to the underlying biology of aging, which includes cellular and molecular hallmarks of aging as well as factors that contribute to age-associated physiological decline and aging phenotypes, such as DNA methylation, plasma proteomics, and elimination of senescent cells [261,262,263,264,265,266].

Biomarker generalizability covers their ability to work across cell types, organs (e.g., kidney, liver, or heart), organ systems (circulatory or immune), species (worms, flies, mice, and humans), and the entire human population [267]. Blood and multi-tissue aging biomarker have been tested and validated in mice using dietary (calorie restriction), genetic (growth hormone receptor knockout), and pharmacological interventions. (e.g., rapamycin) [268,269,270]. Given the significance of cross-species applicability, the American Federation of Aging Research has developed a new definition of a ‘real’ biomarker of aging (such as human–dog dual species and universal pan-mammalian clocks) that works in both human and animal models [30,271,272,273].

Response criteria for biomarkers of aging include responsiveness to both accelerated (e.g., caused by systemic negative effect on longevity like chronic stress or societal factors such as poverty) and decelerated aging (influenced by geroprotectors or lifestyle interventions) [274,275,276,277]. Some interventions and lifestyle, including calorie restriction (CALERIE), exercise trials like DAMA, DO-HEALTH, and Generation-100, mTOR inhibitors, and plasmapheresis, generated sufficient scientific data to support human clinical longevity trials and the potential for biomarker development [277,278,279,280,281,282,283]. There is a relationship between biomarkers of aging and geroprotectors, as one can help validate the other in the context of the examples above. Geroprotectors can be evaluated by their impact on healthspan, lifespan, or assessed biomarkers, while biomarkers of aging can be assessed based on their response to evaluated geroprotectors.

Finally, cost-cutting criteria must be implemented for aging biomarkers in biobanks, clinical trials, and population research. Most established aging biomarkers are too expensive. Recent strategies include probabilistic statistics, low-pass sequencing (like scAge), and targeted high-throughput analysis (e.g., DNAm profiling of specific CpG sites, seen in TIMEseq) [284]. An ideal biomarker represents multiple physiological systems or a combination of biomarkers referred as integrative biomarkers, which examine aging processes and its pace, often assessing biological age compared to chronological age, termed as age acceleration or age deviation [23,285].

Biomarkers also require validation, including biological, cross-species, predictive, analytical, and clinical validation. Biological validation assesses alignment with aging biology, with an emphasis on pathways directly connected to aging rather than associations. Cross-species validation evaluates functionality across species, focusing on phylogenetic conservation [254]. Predictive validation tests biomarker models against independent data for aging-related outcomes. Analytical validation ensures precision and accuracy, crucial for longitudinal aging research [286]. Finally, clinical validation determines if biomarkers accurately identify desired outcomes in human trials [286]. Unlike cross-sectional research, longitudinal studies monitor individuals over time to analyze age-related health outcomes, frequently include genetic data, and assess biomarkers impact on health outcomes [31]. Several global aging cohort studies have been conducted to develop and validate reliable biomarkers, leveraging high-throughput assays to generate multi-omics data. For instance, the National Institute of Aging (NIA) launched the Predictive Biomarkers Initiative to develop high-throughput assays for aging-related processes, improving analytical methods and validating biomarkers across human populations. Similarly, the Aging Biomarker Consortium (ABC) aims to develop reliable biomarkers across multiple organs using multi-omics data, fostering aging cohort studies to understand aging mechanisms and find age-related disorder solutions. Additionally, the MARK-AGE project, funded by the European Commission (EU), aims to conduct a population study with 3200 subjects to identify a set of aging biomarkers, which, when combined following appropriate weighted criteria, should measure biological age better than single markers [287,288]. 

Collectively, these parameters and guidelines, together with the progress in the characterization of the underlying biology of aging, have resulted in the identification of several potential aging biomarkers at the preclinical and clinical level, encompassing the aging hallmarks described above. These include readouts of cellular senescence, proteostasis, and metabolic alterations, as described below.

### 3.1. Cellular Senescence Readouts and Biomarker Candidates

Accumulation of senescent cells and SASP factors in various tissues and organs contributes to aging and age-related diseases. Preclinically, cellular senescence detection methods include Western blotting (WB), immunofluorescence (IF), immunohistochemistry (IHC), flow cytometry, enzyme-linked immunosorbent assay (ELISA), quantitative real-time PCR (qPCR), and liquid chromatography–tandem mass spectrometry (LC MS/MS). Primary senescence markers such as p16^Ink4a^, p21^CIP1^, and SA-β-Gal, as well as secondary markers like SASP and its markers, can be quantified using these biochemical assays in cells, tissues, and mice [113,289,290,291,292,293] (Table 2). As summarized previously, in preclinical studies, the removal of senescent cells and inhibition of SASP through genetic or pharmacological (such as senolytics) approaches can extend health- and lifespan [294,295,296,297,298]. After positive outcomes in preclinical studies, some senolytics have been fast-tracked into human trials to assess their effectiveness and potential side effects, and to assess possible biomarkers of senescence. For instance, in the clinical trials where oral dasatinib with quercetin (D+Q) was dosed in idiopathic pulmonary fibrosis patients, circulating SASP factors were measured in plasma using multiplex discovery platforms; however, the result was inconclusive [299]. In another study (NCT02848131) (Table 1), a three-day D+Q course reduced SA-β-gal-positive (assessed via fluorescence microscopy) and p16^INK4a^ and p21^CIP1^ (via immunohistochemistry) expressing cells in adipose and skin biopsies and lowered SASP factors (including IL-1α, IL-6, and MMP-9/12) in blood samples (measured via multiplex fluorescent bead assay) of diabetic kidney disease patients [58].

These findings demonstrated that senescent cells can be evaluated in clinical samples by measuring SA-β-gal, p16^INK4a^, p21^CIP1^, and SASP. However, currently there are no universally approved biomarkers for senescent cells. Although β-galactosidase activity can be used as a senescence marker, its specificity is limited as it can be identified in other cell types or conditions [300,301,302]. Identification of accurate and precise characterization of SASP and their factors in various physiological and pathological conditions remains challenging, since senescent cells exhibit significant variations in cell types, signaling pathways, and tissue distribution. In summary, advancement in novel techniques and strategies may help to enhance the characterization of senescent cells through more biologically relevant biomarkers. In the future, advancement of artificial intelligence (AI) and deep learning systems will revolutionize the detection of cellular senescence, offering automated and unbiased methods that will enrich histological assessment [303,304,305]. These innovations will help to develop diagnostic and senolytic tools for research and clinical applications.

### 3.2. Autophagy and Protein Quality Control (PQC) Readouts

Autophagy biomarkers are essential to quantify autophagy dynamically in cells and tissues in both physiological and pathological conditions. These biomarkers cover a wide range of degradation mechanisms, including chaperone-mediated autophagy, macroautophagy, microautophagy, and selective subtypes like ferritinophagy, aggrephagy, pexophagy, and mitophagy [306,307,308] (Table 2). Preclinically, autophagy and PQC markers can be detected using biochemical assays such as Western blotting, immunohistochemistry (IHC), immuno-transmission electron microscopy, and immunofluorescence labeling techniques [309]. For instance, CMA activity can be measured via lysosome-associated membrane protein type 2a (LAMP2A) and Hsc70 expression [310,311,312,313,314]. Similarly, protein synthesis rate can be quantified via a reduction in mRNAs encoding ribosomal subunits using RT-qPCR and mass-spectrometry-based proteomics. Translational machinery processes, such as translational output and translation fidelity, can be characterized using ribosomal profiling and luciferase reporter assays [315,316,317,318,319].

Due to positive outcomes from preclinical quantification of autophagy and PQC biomarkers, clinical trials have been initiated to explore the feasibility of using autophagy and PQC readouts as aging biomarkers. For instance, LC3 assessment in leukocytes was used to quantify cellular autophagy after treatment with metformin in a prediabetic and aged cohort (NCT03309007). Although there was no direct measurement of PQC in clinical samples, researchers utilized stable isotope-labeled amino-acids (such as deuterated alanine, phenylalanine, and tyrosine) to assess protein kinetics, including synthesis and breakdown, through gas chromatography–mass spectrometry (GC–MS) (NCT05301179) [320,321]. However, more studies need to be conducted to validate the precise readouts for estimating PQC as well as the underlying biological mechanism for developing aging-related biomarkers. When trying to assess autophagy in clinical samples, several challenge points emerge. Sensitivity issues may develop, especially when detecting low-abundance autophagy markers, which might lead to misinterpretation. Variations in autophagy gene expression across different model systems and tissue types might confound interpretation, limiting their universal application [322,323]. Future improvements, such as functional screening combined with analyses of autophagy markers in body fluids could be potential biomarkers for autophagy [324].

In addition to proteostasis readouts, proteomic platforms have revealed that circulating proteins in body fluids, such as blood, plasma serum, or cerebral spinal fluid (CSF) could serve as potential biomarkers of aging [325,326,327,328,329]. Recent advancements, include the SomaScan assay, a highly multiplexed platform in which plasma was isolated from EDTA-treated blood which led to the identification of proteins whose abundance correlate with age in humans. Proteins such as pleiotrophin (PTN), WNT1-inducible-signaling pathway protein 2 (WISP-2), chordin-like protein 1 (CRDL1), R-spondin-1(RSPO1), EGF-containing fibulin-like extracellular matrix protein 1 (FBLN3), and growth/differentiation factor 15 (MIC-1 and transgelin (TAGL)) have shown positive correlations with age. Conversely, proteins, such as epidermal growth factor receptor (ERBB1), a2-antiplasmin, and A disintegrin and metalloproteinase with thrombospondin motifs 13 (ATS13), exhibit a negative correlation with age. These proteins are also linked to age-related complex diseases, including diabetes, stroke, hypertension, myocardial infarction, and physical measures such as gait speed, grip strength, and frailty [330,331,332,333,334]. Another study employed human plasma proteomics, analyzing proteins such as renin (REN), Uromodulin (UMOD), KLOTHO, Neurexin (NRXN3), complexin 1 (CPLX1), complexin 2 (CPLX2), cardiac myosin light chain (MYL7), and troponin T (TNNT2) from 11 major organs including adipose tissue, arteries, brain, heart, immune tissue, intestine, kidney, liver, lungs, muscle, and pancreas that were differently enriched with age across the cohort. Moreover, the integration of plasma proteomics with machine learning has given rise to organ aging models that estimate organ-specific biological ages [335,336,337,338,339,340,341,342,343]. Although these results offer valuable insights into the proteins and pathways associated with aging and suggests that the plasma proteomic profile may be a potential biomarker of aging, there are still limitations that need to be considered; these include cross-reactivity in detection technologies, relative protein concentration measurements, and incomplete proteomic coverage, which indicates that many age-associated proteins and their implications in the aging process remain to be discovered. Additionally, broader validation across diverse populations is needed to confirm these findings. Future research should focus on expanding and refining proteomic analysis and applying advanced machine learning and AI to improve the precision of age-related health assessments, which is essential for progress in anti-aging treatments and personalized medicine [335,344,345].
ijms-25-06793-t002_Table 2Table 2Aging biomarker candidates in the preclinical and clinical space.TypeProcesses/MaterialBiomarkersReferencesMacroautophagyAutophagosome formationAtg5-Atg12, Atg16L, LC3Atg9, BECN1/Beclin1/Vps30/Atg6,Atg14/Bakor, DRAM1, ZFYVE1/DFCP1[346,347,348]Substrate of macroautophagyp62[349]Chaperone-mediated autophagyKFERQ-like amino acid sequence recognized by HSC70LAMP-2A, HSC70[350]Microautophagy Multi-vesicular bodiesRab7, Vac8, Atg18, ESCRT(VPS4)[351]Lysosome Autophagosome content degraded by acid hydrolytic enzymesLAMP-1, LAMP-2[352]Selective autophagyMitophagyPINK/Parkin, BNIP3, NIX/BNIP3L, Atg32[353,354,355]Aggrephagyp62/SQSTM1[356]Mitochondrial homeostasisFatty acid metabolismFatty acids, NAD^+^[357]Cellular senescenceCell-cycle arrestp53, p16, p21, Ki67[358]Structural changeSA-b -gal, SAHFs[358,359]Secondary markersSASP chemokine (IL-6, IL-7, IL-15),Cytokines (e.g., IL-8, CCL3, CCL4),Growth factors (e.g., GDF-15, activin A)[299,360]Protein quality controlProtein kineticsIsotope-labeled amino-acids (such as deuterated alanine, phenylalanine, and tyrosine)[320,321]Circulating proteinsBlood/Plasma/SerumPTN, WNT1, WISP-2, CRDL1, RSPO1, FBLN3, GDF15, TAGL, a-Klotho, TGF-b, acyl-carnitine, GDF15, RAGE, VEGFA, PARC, MMP2[330,361,362,363]MetabolitesLipids/FAHDL, LDL, VLDL, TG, PUFA, omega-3 FAs, EPA, DHA[364,365,366]HormonesDHEA-S, androgens, progestins, and pregnenolones[367,368]Amino acidTyrosine increased;Tryptophan tends to decrease[369,370]Inflammatory markersCRP, TNF-alpha, IL-6, IL-8, IL1-b, IL-15[371,372]


### 3.3. Metabolites Readouts and Biomarkers Candidates

Changes in metabolic activity, metabolite levels, and fluxes with aging have been extensively studied, encompassing both preclinical and clinical studies (Table 2). These have been observed in alterations of lipid and lipoprotein metabolism, steroid hormone biosynthesis, amino acid turnover, mitochondrial metabolites, inflammation, and NAD^+^ levels. Variations in the levels and fluxes of specific metabolites from these pathways can serve as biomarkers for aging [373,374,375,376,377,378,379]. Analytical techniques, including nuclear magnetic resonance (NMR) and mass spectrometry (MS), have been used to detect and quantify these age-related metabolites. For instance, NMR studies have shown that the lipoprotein profile changes with age, with decreased high-density lipoprotein (HDL) and increased very-low-density lipoprotein (VLDL), low-density lipoprotein (LDL), triglycerides (TGs), cholesterol, polyunsaturated fatty acids (PUFAs), and fatty acids (FAs), both in blood and urine [364,365]. The aging process also impacts omega-3 FAs fatty acids such as eicosapentaenoic acid (EPA) and docosahexaenoic acid (DHA), decreased in red blood cells of aged subjects, analyzed using the Omega-3 index^®^ methodology and gas chromatography [366]. Besides, aging also correlates with increased levels of acyl-carnitine in the blood and plasma, a molecule that facilitates the transport of long-chain fatty acids into mitochondria [362,380,381,382]. Interestingly, supplementation of urolithin A (UA) has been observed to reduce plasma acylcarnitine levels, which is associated with improved mitochondrial efficiency (Table 1). This suggests that FAs and acylcarnitine levels in body fluids could serve as potential biomarkers for aging and its related interventions [85,86,383]. Moreover, age-related declines in various steroid hormones, including dehydroepiandrosterone sulfate (DHEA-S), androgens, progestins, and pregnenolones, have been observed in human cohorts, highlighting their potential as biomarkers for metabolic changes occurring in aging [367,368]. In terms of amino acid levels, the research thus far has revealed a complex pattern of changes. In serum, tryptophan, threonine, serine, methionine, and cysteine have been observed to decrease with age, while tyrosine increased [369]. Another group reported that in both plasma and serum tryptophan tends to decrease [369,384,385,386], while tyrosine increases with age [369,384,387]. Although partially replicated, these trends are not uniform and might be affected by factors such as sample size and biological source. Additionally, other metabolites, related to amino acids and mitochondrial metabolism such as citrulline, carnitine, and glutamate have been connected to aging in saliva of 27 healthy volunteers in Okinawa, proposing their use as effective biomarkers for aging [379,388]. Furthermore, there is significant increase in inflammatory markers during aging, including cytokines (IL-6, IL-10, and TNF-α), interferons alpha and beta (IFN-a and IFN-b), and other indicator like C-reactive protein (CRP) in plasma, quantifiable with ELISA or other platforms [389,390,391]. Interestingly, interventions that enhance mitochondrial efficiency such as UA also resulted in lowering CRP levels in a human plasma cohort, suggesting the potential of CRP as an effective biomarker of aging and interventions affecting metabolic pathways [86,392]. Another metabolite which is being actively investigated for its links with aging, and that has been discussed also as an intervention target in the above chapters, is NAD^+^. In preclinical studies, supplementation with NAD^+^ precursors along with lifestyle interventions can potentially extend lifespan and enhance physiological functions [97,393,394,395,396]. Several clinical studies have shown that in both healthy and diseased individuals, supplementation with NAD^+^ precursors (NR, NAM, NMN, and NA) leads to measurable changes in NAD^+^ and related metabolites such as nicotinic acid dinucleotide (NAAD) and methyl-NAM. These readouts may serve as markers to assess the impact of dietary supplementation and overall health status [99,100,357,397,398,399,400,401,402]. Therefore, quantification of NAD^+^ and of its related metabolites is of critical relevance for possible biomarker development. NAD^+^ can be quantified from muscle biopsies, red blood cells, and saliva through biochemical assays (e.g., enzymatic and calorimetric assay), which are highly sensitive and affordable [388,402,403,404]. These methods are, however, limited only to NAD^+^ detection. Several methods are available for detecting trace levels of NAD^+^ and its metabolome while retaining accuracy, including capillary electrophoresis, NMR, HPLC-NMR, and UV-detection HPLC. Additionally, researchers have developed multiple reaction monitoring (MRM)-based techniques to improve detection accuracy [405,406,407,408,409,410,411,412,413,414,415,416]. Despite advancements in NAD^+^ metabolome measurement technologies, variability in chromatographic analysis can lead to significant fluctuations in NAD^+^ concentrations in tissues or body fluids, often resulting in levels that are below the detection threshold.

In conclusion, metabolite measurements are emerging as promising biomarkers for aging, though their utilization encounters significant challenges and limitations. Metabolite identification is an ever-evolving field, with new reference libraries and methods adding complexity to data analysis. Confounding variables such as demographics and lifestyle also complicate accurate metabolite quantification. The lack of long-term data is a significant barrier for biomarker development, addressable through detailed longitudinal studies [387,417,418,419,420]. Future efforts must focus on solidifying study designs and data collection to validate metabolic readouts as effective biomarkers for aging and interventions efficacy.

## 4. Biological Aging Clocks

Most animals, including humans, experience biological aging, causing biological decline that is accompanied by an exponential increase in mortality with time (age). This relationship between age and mortality can be expressed with a Gompertz mortality distribution [421]. Adherence to Gompertz mortality is a defining quality of what we mean when we say that a species “ages”. Biological age is then defined as the time constant in the Gompertz distribution and, on an individual level, is not the same as chronological age. On a cohort level, Gompertz mortality predicts the fraction of animals who are going to die at a given age, and biological age is identical to chronological age. However, individual biological aging trajectories are impacted by lifestyle, genetics, and environmental factors, meaning that individuals at the same chronological age can vary substantially in their mortality and disease risk. On an individual level, true biological age is therefore not identical to simple chronological age, and this is the fundamental reason for the interest in biological aging clocks. A true biological aging clock is a computational algorithm that, based on molecular, clinical, and demographic parameters, assigns an accurate biological age to individual subjects. Attempts to construct biological aging clocks to determine biological age from observable biomarkers including physical features have a long history [422,423,424]. Biological aging clocks can be constructed based on a wide range of biological features/biomarkers as described above, and selected examples are summarized in Table 3. For recent exhaustive reviews on currently available biological aging clocks, see Moqri et al. and Bafei and Shen [285,425].

### 4.1. Evolution of Biological Aging Clocks and Current Limitations

In addition to the biological feature space used, biological aging clocks may differ in “age” that they aim to predict. As outlined above, conceptually, true biological age is best defined as a Gompertz age, that is, the age commensurate with an individual’s future risk of dying from all intrinsic causes. Gompertz biological age is the key determinant of an individual’s risk of age-dependent disease and all-cause mortality. However, historically, many biological aging clocks use different operational definitions, for example, defining biological age as the age at which the test subject’s physiology (as determined by its position in feature space) would be approximately normal for members of a reference cohort [428,429]. First-generation DNA methylation (DNAm) clocks, for example, follow this approach [23,25]. While first-generation DNAm clocks have attained impressive accuracy in determining chronological age, they are generally less powerful predictors of future morbidity and mortality [27,430].

Recently, second-generation clocks have been constructed, aiming to predict future mortality more directly from biological parameters [26,27,431,432,433]. Second-generation clocks therefore attempt to determine true Gompertz biological age [27]. The second-generation clocks share some similarities with traditional clinical risk markers, such as the Atherosclerosis Cardiovascular Disease (ASCVD) score [434], but differ in that they attempt to predict all-cause mortality, better reflecting the high degree of interconnectivity between organ system and disease etiology [16,26,27,35,431,432,433]. This is important because healthy aging is more than simply the absence of specific diseases, and, unlike existing clinical risk markers, knowledge of true biological age allows identification of individuals likely to remain free from age-dependent morbidity and mortality for years to come, thereby providing normative targets for clinical intervention and individual guidance for the promotion of healthy aging.

However, construction of second-generation clocks requires historic -omics data (or appropriate bio-banked samples) from large cohorts of subjects for whom decades of disease and mortality follow-up have also been collected. Unfortunately, due to the dearth of large -omics datasets with such long mortality follow-up, very few second-generation -omics clocks have been constructed to date. Even those that have been constructed typically employ indirect methods, aiming to predict clinical biomarkers with known impact on mortality from -omics features [27,28,29]. However, as large datasets with long mortality follow-up time and -omics data gradually become available, the construction of next generation -omics clocks able to predict true Gompertz biological age will soon become a reality.

Fortunately, unlike for most types of -omics data, large datasets containing standard clinical chemistry (e.g., complete blood count, low-density lipoprotein, and glycohemoglobin) and physiological features (e.g., blood pressure, heart rate, and body mass index) are available for some cohorts with long mortality follow-up (examples include the US National Health and Nutrition Examination Survey (NHANES) III, NHANES IV, and UK Biobank). Such datasets have enabled the generation of “clinical clocks”, which aim to predict future all-cause mortality and morbidity directly from clinical features and biomarkers [7,26,27,431,435,436,437]. Because clinical features typically have intrinsic well-established biological and pathophysiological meaning, findings from clinical clocks are comparatively easier to interpret and act upon. Clinical clocks are therefore better suited to detect hidden or subclinical diseases and inform primordial prevention—a concept that involves preventing the development of clinical risk factors prior to the onset of full-blown disease and which therefore precedes primary prevention [438,439]. For example, utilizing the NHANES IV cohort with 20-year mortality follow-up, we recently showed that our clinical aging clock, PCAge, trained against all-cause mortality, could identify metabolic dysfunction and cardiac and renal dysfunction, as well as markers of inflammation as predictors of accelerated biological aging [21]. A reduced clinical clock, LinAge, based directly on PCAge, has equivalent predictive power but uses significantly fewer features [21]. Such clinical clocks are practically optimized for rapid translation into clinical practice after their validation. For the time being, current and future clinical clocks remain more immediately clinically actionable than -omics clocks, both current and under development.

However, all current biological aging clocks systematically convolute intrinsic biological aging with disease signatures [440,441]. Because age is intrinsically and strongly correlated with age-dependent diseases, this is a fundamental problem when constructing biological aging clocks based on data derived from cohorts of aging animals. None of the currently employed linear techniques can separate intrinsic aging signatures from those of age-dependent diseases. Certainly, for clinical clocks trained to predict future mortality, some of the predictive power derives from identifying hidden and subclinical disease signatures or disease risk markers. While early detection of hidden or subclinical diseases is useful, it is important to distinguish such disease signatures from systemic aging. Most current biological aging clocks cannot be interpreted to deliver clear mechanistic insights on the biology of aging and cannot delineate causes from consequences of aging and of age-dependent diseases. Given the need for clinically actionable risk–benefit and efficacy evaluation of novel pre-emptive and preventative healthspan interventions, there is a crucial unmet clinical need for a qualitatively different type of biological aging clock.

### 4.2. The Need for a New Generation of Biological Aging Clocks

When developing and testing interventions to maximize human healthspan, a critical problem is clinical equipoise, that is, the accurate evaluation of the risks associated with any intervention relative to its potential or actual benefit. Because of the exponential nature of age-dependent mortality increase, any intervention that demonstrably slows intrinsic aging would have disproportional beneficial impact, especially when applied early in life. Tools to accurately quantify changes in true Gompertz biological age are therefore essential for the development and validation of any healthspan-optimizing diet, lifestyle, supplement, geroprotector, or drug intervention. A true biological aging clock should predict disease-specific and all-cause mortality in individuals in a way analogous to current cohort-level risk markers.

To date, no such clocks exist. While many different biological aging clocks have been developed, typically by applying machine learning tools and linear models to clinical and -omics data, all current biological aging clocks suffer from substantial flaws limiting their clinical utility. Next generation clocks therefore need to serve three related purposes. First, they need to be able to readily detect disease substantially earlier (the “hidden sickness”) than the best currently available diagnostic approaches. Second, in addition to detecting subclinical disease early, such clocks need to be sensitive to individual differences in disease risk and quantify intrinsic biological resilience. Third, utilizing aging biomarkers, aging clocks need to provide actionable insights into mechanisms of aging and to facilitate interventions against them. Only fully developed aging clocks will allow healthcare providers and governments to navigate the complexities of the risk–benefit analysis required to add years to healthy lifespan by intervening decades before disease onset.

## 5. Discussion

Chronological age is the amount of time elapsed from birth, whereas biological age is a reflection of the decline in organismal resilience and damage accumulated over time (Figure 2) [442]. Biological age is a valuable metric as it estimates age and lifestyle-related loss of function of organs, allowing prescription of optimized interventions. Aging is a complex multifactorial process and from the biological viewpoint can be attempted to be simplified to hallmarks including cellular senescence, autophagy, PQC, and mitochondria [1,8]. Interventions have been designed for targeting components of pathways underlying these hallmarks, with the aim to alleviate the burden of biological aging, as progress in the preclinical space indicates. The hallmarks are intricately intertwined, for example dysfunction in mitochondria contribute to cellular senescence or impairment of autophagy and PQC [443,444,445]. This, in turn, has resulted in the discovery of interventions that show benefits in numerous hallmarks, especially the pleiotropic geroprotectors rapamycin and metformin, which are already widely used clinically [66,67,68,69,77,78]. Notably, in this review, a common aspect to the hallmarks of aging is oxidative stress, supporting the free radical theory of aging in which accumulation of free radicals, particularly reactive oxygen species generated by the mitochondria, cause macromolecular damage such as DNA mutations during aging [446]. Hence, it is unsurprising that antioxidants such as quercetin and, in particular, mitochondrial antioxidants such as MitoQ have positive outcomes preclinically and clinically [447]. Inflammaging is also an impactful and integrative hallmark that is targeted by senolytics and geroprotectors such as rapamycin and rapalogs [53].

However, the development of approaches that can impact specific mechanisms of aging will be critical to generate more precise and effective interventions in this field. Novel targeting of the aging hallmarks includes a variety of approaches such as engineered mitochondrial nucleases that remove mtDNA mutations, HSP inhibitors or activators to restore proteostasis, senomorphics, NAD^+^ precursors, mitochondrial modulators, and previously uncharacterized natural molecules able to selectively target aging pathways, as summarized in the chapters above. Critical to the translational applications of all these interventions will not only be their potential efficacy, but the possibility to establish solid clinical readouts or biomarkers. Some longevity approaches such as stem-cell or microbiome therapies were not discussed in this review despite recent advancements. For example, mesenchymal stem cell therapy improved fitness and immunity of patients with frailty [448], gut microbiome clocks are an exciting new field [449], and probiotics are a popular supplement in the elderly with some clinical improvements in immunity [450]. However, the biomarker readouts for such therapies are not specifically defined and are not necessarily related to aging, and, in fact, most of the readouts for the clinical trials in these fields are regarding immunity [448,450]. Developing biomarkers of aging into clinically useful tools has significant promise for improving healthcare and extending life expectancy, thereby lowering healthcare costs as well as encouraging public wellness. For accurate age-related changes, assessment, and health outcomes prediction, biomarkers that capture the temporal dynamics of aging and distinguish between chronological and biological age are important.

Based on this possibility, worldwide initiatives are on the way to identify clinical biomarkers of aging. These include the Longevity Genomics Program (LGP), the Human Longevity Project, and the European Research Initiative for Healthy Aging (ERA-AGE), which use strategies ranging from identifying genetic factors linked to healthy aging and longevity to the investigation of lifestyle and environmental factors of long-lived individuals and the characterization of aging biology to identify clinical biomarkers that can predict healthy aging and age-related diseases [451,452,453]. Similarly, The Targeting Aging with Metformin (TAME) project, led by the American Federation for Aging Research (AFAR), aims to identify biomarkers of aging and assess the potential of metformin to delay age-related diseases and extend healthspan [454,455].

Despite these advances, the dynamic nature of aging makes it difficult to find stable biomarkers that can track its rate over time. The primary challenge is the lack of a consistent approach for determining a definitive “ground truth” for aging, which restricts the validation of biomarkers [27,456,457,458,459]. Furthermore, it is still uncertain which data sources most accurately reflect biological age [363,460,461]. Another issue is the inherent bias towards improving the accuracy of predictive models, which may conflict with the true representation of aging rates—a paradox that complicates the quest for precise biomarkers [462]. Lastly, the clinical relevance of various aging biomarkers is yet to be fully understood, necessitating extensive longitudinal research to ascertain their utility.

Consequential to this is the inherent limitations that current biological aging clocks, stemming from biomarker patterns, still carry, that need to be addressed before they can be considered ready for wide clinical implementation. Firstly, most, if not all, of the clocks developed thus far (Table 3) do not agree with one another. This is evident, for instance, when observing the age deltas or residuals of the clocks, which mostly differ amongst all of these tools. Secondly, -omics and methylation-based clocks are not immediately interpretable biologically and therefore not actionable in a clinical setting to provide a rapid and precise recommendation. Thirdly, many of the current clocks are trained on and largely predict chronological age. Yet, a true aging clock should predict all-cause mortality, independently of any existing subclinical or clinical disease, utilizing a panel of aging biomarkers. This ideal clock will not require knowledge of chronological age and yet demonstrate that the risk of dying follows a Gompertz-like distribution. For clocks to warrant clinical use, the biological age predicted by these algorithms will need to converge with each other and predict historic mortality in multiple external cohorts, allowing validation studies to compare their accuracy and precision. Addressing these challenges is essential for the advancement of biomarkers and biological aging clocks that could reliably indicate the aging pace and the best possible interventions to modulate it.

In conclusion, the development of robust biomarkers for aging and subsequential aging clocks requires thorough validation to ascertain their clinical relevance. Biomarkers of aging have the potential to help in developing human longevity therapies and personalized healthcare decision-making. This necessitates the advancement of targeted longevity therapeutics to target the thus-far established biological pathways of aging.

## Figures and Tables

**Figure 1 ijms-25-06793-f001:**
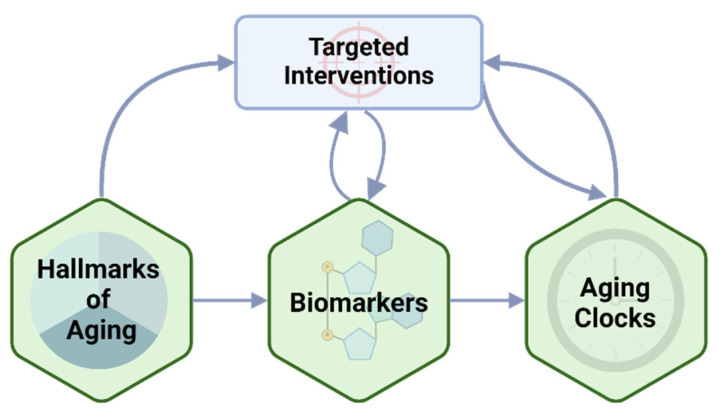
The hallmarks of aging provide a foundation for developing targets of aging and biomarkers for the detection of age-related diseases. Aging clocks can be developed from biomarkers for the detection and quantification of biological age. Efficiency of target intervention can validate and be validated using biomarkers and aging clocks. Created with BioRender.com.

**Figure 2 ijms-25-06793-f002:**
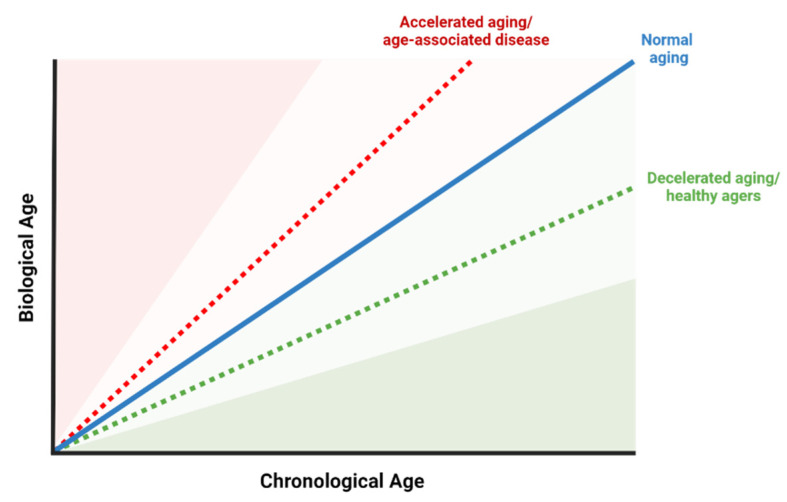
Chronological age is the time lived after birth, whereas biological age reflects the health condition of cells and tissues. Accelerated aging, in which biological age is higher than chronological age, increases risk of age-related disease and mortality. Decelerated aging is favorable as damage to systemic circulation or organs is reduced along with risk of age-related disease. Created with BioRender.com.

**Table 1 ijms-25-06793-t001:** Compounds targeting hallmarks of aging under clinical investigation or in advanced stages of preclinical research.

Compound	Current Use (If Any)	Aging Hallmark Target	Preclinical Research Outcome	Clinical Trial Outcome	References	Year of Publication
Dasatinib and quercetin	Dasatinib: tyrosine kinase inhibitor for leukemia; Quercetin: senolytic supplement	Senescence	Reduced senescence in adipose tissue and spinal discs but not in liver and muscle of old mice	Reduced p16 and p21 expression in skin; Reduced SASP; No biological aging with second and third generation aging clocks	[58,59,60,61]	(2019, 2024, 2023, 2021)
Fisetin	Senolytic supplement	Senescence	Reduction in senescence in animal models	No clinical trials	[62,63,64]	(2024, 2018, 2023)
Ruxolitinib	FDA-approved JAK inhibitor for myelofibrosis	Senescence	Decrease in inflammatory cytokines in mice	No clinical trials	[65]	(2015)
Rapamycin	FDA-approved immuno-suppressant	Senescence; Mitochondria	Reduced mtDNA mutations and oxidative stress in mousce liver; Increased autophagy in mouse kidney	Reduction in p16 expression in skin after topical application	[66,67,68,69]	(2019, 2017, 2016, 2023)
Cycloastragenol (TA-65)	Telomerase activator supplement	Senescence	Activates telomerase reverse transcriptase in primary neurons; Reduces senescence and telomere attrition in spinal disc cells	Increased telomere length; Improvement in immunity	[70,71,72,73]	(2016, 2023, 2021, 2014)
Spermidine	Supplement	Autophagy	Induces autophagy and reduces oxidative stress in brain and arteries of aging mice	Improvement in memory (short term treatment); No change in memory (long term treatment)	[74,75,76]	(2018, 2021, 2022)
Metformin	Energy sensing and metabolism regulation	Autophagy; Mitochondrial biogenesis	Increased autophagy ex vivo in aged human T cells	Improvement in mitochondrial pathways	[77,78]	(2020, 2018)
Monorden and tanespimycin	Monorden: natural product; Tanespimycin: antibiotic	PQC (Hsp90 inhibitor)	Increase in lifespan of *C. elegans*	No clinical data	[79]	(2019)
Phenothiazines	Antipsychotic drugs	PQC (proteasome activator)	Increase in lifespan of *C. elegans*	No clinical data	[80]	(2021)
Omega-3	Supplement	PQC (protein synthesis)	Inhibits proteasome activity in HEK293 cells; Induces autophagy in human retinal cells	Increased protein synthesis and decreased protein degradation in muscle; Improved muscle function	[81,82,83]	(2011, 2016, 2020)
Urolithin A	Supplement	Mitophagy	Increased lifespan, mitophagy, and mitochondrial function and decreased oxidative stress in *C. elegans;* Increased mitophagy in mice	Increased mitochondrial biogenesis; Improved expression of mitochondrial genesets; Increased mitophagy and muscle strength	[84,85,86]	(2016,2019, 2022)
CoQ10 and MitoQ	Supplement	Mitochondrial function and antioxidant	Increase in mitochondrial activity in mouse brain and oocytes; Increase in fertility of aged mice; No change in mortality of mice	Increased skin elasticity and collagen synthesis; Improved vascular function	[87,88]	(2023, 2018)
MitoTALENS	Candidate therapeutic	Mitochondria	Reduction in mtDNA mutations	No clinical data	[89]	(2020)
Epicatechin	Supplement	Mitochondrial biogenesis	Increases lifespan in aged mice; Decrease in muscle degeneration in aged mice	Improved mitochondrial structure	[90]	(2021)
Sulforophane	Supplement	Mitochondria	Improved mitochondrial function in rats	No clinical data	[91,92]	(2022, 2020)
Nicotinamide riboside	Supplement	NAD^+^	Increased lifespan and mitochondrial function and reduced cell senescence in mice	Mild improvements in physical performance in elderlies; Amelioration of inflammation and mitochondrial gene expression; No improvement in cognitive function	[93,94,95,96,97]	(2020, 2019, 2024, 2018, 2016)
Nicotinamide mononucleotide	Supplement	NAD^+^	Reduced protein aggregation and improved mitochondrial structure in mice	Improved insulin sensitivity in pre-diabetic women; Improved physical fitness and high density lipoprotein levels; No improvement in insulin resistance and amelioration of increase in biological age in mixed gender trial	[98,99,100,101,102,103]	(2022, 2023, 2022, 2021, 2022, 2022)
Nicotinic acid	Supplement	NAD^+^	Increases mitochondrial biogenesis and decreases autophagy in cancer cachexia mouse model	Reduced triglyceride and total cholesterol levels; Improved memory and insulin sensitivity	[104,105]	(2006, 1985)
Trigonelline	Supplement	NAD^+^	Increases lifespan in *C. elegans*; Enhances mitochondrial and muscle function in aged mice	No clinical data	[106]	(2024)
NTX-748	Candidate therapeutic	NAD^+^	Increase in NAD^+^ levels in spleen and liver of mice	No clinical data	[107]	(2023)

**Table 3 ijms-25-06793-t003:** Examples of biological aging clocks.

Input Variable	Training Data
Clinical blood markers	Chronological age: Nakamura et al. [11], Tian et al. [16], BloodAge [22] Mortality: PhenoAge [27], ENABL Age [20], PCAge/LinAge [21], Bortz et al. [426]
Physiological parameters	Chronological age: Pyrkov et al. [15]
DNA methylation	Chronological age: Hannum [25], Horvath [23] Mortality: PhenoAge * [27], GrimAge ** (versions 1 and 2) [28,29] Rate of aging: DunedinPoAm * [24], DunedinPACE * [260]
-omics data	Chronological age: Fleischer et al. (transcriptomic markers) [427], Oh et al. (proteomics markers) [335], Metabolomic age (metabolomics markers) [33], iAge (immune protein markers) [36], GlycanAge (glycobiology markers) [37]

* Indirectly trained on clinical blood markers. ** Indirectly trained on plasma proteins.

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
