# Peer review of "Towards Healthy Longevity: Comprehensive Insights from Molecular Targets and Biomarkers to Biological Clocks"

_ijms, 2024, doi:10.3390/ijms25126793_

Round 1

Reviewer 1 Report

Comments and Suggestions for Authors

The authors wrote a comprehensive review describing the role of anti-aging drugs and supplements. The part about biological clocks is very nice to read.

I have only one concern:

 The impact of inflammation on aging process should be described with more details in the introduction, alongside with its interplay with oxidative stress which has not been discussed. Probably it would be worth mentioning something about the “free-radical theory of aging”.

In addition, the concept of “inflammaging” should be discussed.  

Author Response

We appreciate the Reviewer’s comments related to including the relevance of inflammation and oxidative stress in the context of aging. We have added these concepts in the introduction, and briefly reported on their modulation by geroprotectors and other metabolism-modulating interventions; see highlighted changes in the manuscript in the introduction and discussion sections; we also included relevant references, and we thank the reviewer for her/his comment.

Please also see the attachment 

Reviewer 2 Report

Comments and Suggestions for Authors

Yusri and Kumar et al. comprehensively reviewed the molecular mechanisms of aging, potential therapeutic targets, and biomarkers used to assess biological age. They discussed hallmarks of aging, including genomic instability, telomere attrition, epigenetic alterations, and mitochondrial dysfunction. Additionally, the authors introduced current anti-aging interventions and highlighted some recent progress in clinical trials. Furthermore, they discussed the development and utility of biological aging clocks, which integrate various biomarkers to estimate biological age and predict age-related diseases. I have the following suggestions:

Major:

1.       While the title is about aging, the paper focused heavily on anti-aging interventions and aging biomarkers. Some major age-associated diseases such as atherosclerosis, dementia, and osteoporosis are not covered in the review. Please consider revising the title if anti-aging is the focus. Otherwise, additional sections should be used to discuss these age-related conditions.

Minor:

2.       For Table I, it would be helpful to add an additional column to list the year of the corresponding publications.

3.       In Section 2.1, "There are two primary types..." could be confusing. The authors may consider revising the structure as: "There are two primary types: replicative senescence, caused by..., and stress-induced premature senescence, triggered by various stressors, including...," for example.

Author Response

Major:

  1. While the title is about aging, the paper focused heavily on anti-aging interventions and aging biomarkers. Some major age-associated diseases such as atherosclerosis, dementia, and osteoporosis are not covered in the review. Please consider revising the title if anti-aging is the focus. Otherwise, additional sections should be used to discuss these age-related conditions.

We thank the Reviewer for her/his critical evaluation of our manuscript. This review intends to focus on the current knowledge of the biology of the aging process, and of interventions and developments in the biomarkers or clocks space that would promote and allow to measure a compression of age-associated morbidities (healthy longevity), rather than describing age-associated disorders specifically.

Therefore, to incorporate the Reviewer’s feedback and better align title and content of our manuscript, we propose the updated title: “Towards Healthy Longevity: Comprehensive Insights from Molecular Targets and Biomarkers to Biological Clocks”. Accordingly, we have also updated the abstract and keywords, as highlighted in the main text, to further emphasize the relevance of establishing healthy longevity strategies.

We hope to have answered the Reviewer’s comment, and we thank the Reviewer again for her/his valuable suggestion to improve our manuscript.

Minor:

  1. For Table I, it would be helpful to add an additional column to list the year of the corresponding publications.

We thank the Reviewer for this comment, and have updated the table accordingly.

  1. In Section 2.1, "There are two primary types..." could be confusing. The authors may consider revising the structure as: "There are two primary types: replicative senescence, caused by..., and stress-induced premature senescence, triggered by various stressors, including...," for example.

We have updated the main text accordingly (see highlighted text in the main manuscript file), and we thank the reviewer again for her/his input.

Please see also the attachment
